# An Improved Unauthorized Unmanned Aerial Vehicle Detection Algorithm Using Radiofrequency-Based Statistical Fingerprint Analysis

**DOI:** 10.3390/s19020274

**Published:** 2019-01-11

**Authors:** Shengying Yang, Huibin Qin, Xiaolin Liang, Thomas Aaron Gulliver

**Affiliations:** 1Institute of Electron Device & Application, Hangzhou Dianzi University, Hangzhou 310018, China; ysyhdu@126.com; 2Science and Technology on Electronic Test & Measurement Laboratory, The 41st Research Institute of CETC, Qingdao 266555, China; iamxiaolin2016@126.com; 3Department of Electrical Computer Engineering, University of Victoria, Victoria, BC V8W 2Y2, Canada; agullive@ece.uvic.ca

**Keywords:** spectrum sensing, radio frequency (RF), singular value decomposition (SVD), spectrum accumulation (SA), statistical fingerprint analysis (SFA)

## Abstract

Unmanned aerial vehicles (UAVs) are now readily available worldwide and users can easily fly them remotely using smart controllers. This has created the problem of keeping unauthorized UAVs away from private or sensitive areas where they can be a personal or public threat. This paper proposes an improved radio frequency (RF)-based method to detect UAVs. The clutter (interference) is eliminated using a background filtering method. Then singular value decomposition (SVD) and average filtering are used to reduce the noise and improve the signal to noise ratio (SNR). Spectrum accumulation (SA) and statistical fingerprint analysis (SFA) are employed to provide two frequency estimates. These estimates are used to determine if a UAV is present in the detection environment. The data size is reduced using a region of interest (ROI), and this improves the system efficiency and improves azimuth estimation accuracy. Detection results are obtained using real UAV RF signals obtained experimentally which show that the proposed method is more effective than other well-known detection algorithms. The recognition rate with this method is close to 100% within a distance of 2.4 km and greater than 90% within a distance of 3 km. Further, multiple UAVs can be detected accurately using the proposed method.

## 1. Introduction

The Internet of Things (IoT) has attracted significant attention worldwide due to the evolution of pervasive smart environments which can connect people and things anytime, anywhere, and anyplace. The IoT is suitable for many applications such as smart environments [1], remote sensing [2], public security [3], smart traffic [4], health care [5], intelligent cities [6], emergency services [7], and industrial control [8]. Unmanned aerial vehicles (UAVs) were first employed by the military [9,10,11,12]. Since then, many civilian applications have been developed such as UAV photogrammetry which can provide high spatial resolution data due to the low operational altitude [13,14,15]. The relatively low cost of UAV platforms has led to their wide popularity in many fields. UAVs have been employed in ecologic observation [16], vegetation monitoring [17], water quality monitoring [18], precision agriculture [19], surveying [20], natural resource assessment [21], avalanche monitoring [22], forest inventories [23], wildlife management [24], and coastal environment monitoring [25]. Further, they can complement or replace traditional satellite-based methods [26].

The pervasive use of UAVs has led to technical and societal concerns related to security, privacy, and public safety which must be addressed. For example, a UAV interrupted a US Open tennis match and another crashed at the White House. Shooting down the UAV is the most direct solution, but this is typically illegal. In [27], a genetic algorithm (GA) was employed to jam UAV signals. In [28], GPS signals were jammed to disable a UAV. These methods can be improved if the UAV is first detected and located. Several algorithms have been developed which can be used for UAV detection. These algorithms are based on video, sound, radar, temperature, radio frequency (RF), and WiFi [29,30,31,32,33,34]. Active radar was used to detect UAVs in [29]. However, this method can only provide accurate detection results over short distances. A UAV was detected using a calibrated radar cross section (RCS) technique in [30], but the RCS must be known a priori and this can be difficult to acquire. In [31], UAV detection was achieved with a complex-log Fourier transform method and spectrogram features [31]. However, data is needed over a long time period which makes this approach complex and slow. Log-polar transformations and space-variant resolution were used in [32] for UAV detection but the accuracy is poor over long distances. A UAV was located in [33] using an acoustic array at an effective range of 300 m. A multiple input multiple output (MIMO) radar was used to detect and track a UAV in [34]. However, this radar is complex and has poor detection performance due to low received power and receiver sensitivity.

An artificial neural network (ANN) was used in [35] to detect a UAV by analyzing the RF signal characteristics, but this approach cannot provide azimuth information. A passive radar using digital audio broadcasting (DAB) was proposed in [36] to detect a UAV. However, this method can only determine if a UAV is present over short distances and cannot provide both azimuth and range estimates. Range-velocity processing was used in [37] with a MIMO radar in the Ka band for UAV detection [37], but this system is only effective over distances of several hundred meters. Multiple heterogeneous sensors were used to track and detect a UAV in [38]. However, significant time is required for track extraction and multi-target detection. A distributed passive radar was developed in [39] for UAV detection.

UAV detection techniques based on video have been developed. However, it is challenging to distinguish a UAV from other flying objects [40]. In [41], an audio classification system using data mining techniques was proposed for UAV detection, but this system can only work well in quiet environments. In [42], UAV detection was considered using a radar sensor which employs characteristic features, but this system was only evaluated in a laboratory environment. A temperature-based method was proposed to detect a UAV in [43] which is only effective for fixed-wing devices. A method was proposed to detect UAV communication signals in [44]. A vision-based algorithm was proposed in [45] to detect UAVs and estimate their distance. However, this approach is limited to short distances less than 25 m. In [46], a tracking algorithm was developed to detect a moving UAV. Image processing was considered in [47] to identify multiple UAVs [47], but this technique is complex.

A typical UAV has a very small RCS which make radar-based detection algorithms unsuitable for detection [48]. RF-based methods are more appropriate as they can achieve a higher processing gain. In this paper, a spectrum sensing system is employed to detect a UAV. The contributions are as follows:(1)Spectrum accumulation (SA) and statistical fingerprint analysis (SFA) techniques are used to provide frequency estimates of RF signals. These estimates are used to determine if a UAV is present in the detection environment.(2)A region of interest (ROI) is defined to reduce the data size, improve the system efficiency and provide accurate azimuth estimates.(3)The performance of the proposed algorithm is compared with that using several well-known techniques in the literature. Further, the ability to detect multiple UAVs with the proposed algorithm is evaluated.

The remainder of this paper is organized as follows: the system model is given in Section 2. Section 3 presents the proposed detection algorithm and the detection results are given in Section 4. Finally, Section 5 concludes the paper.

## 2. System Model

A UAV communicates with the ground controller using RF signals so these signals can be used for detection. Orthogonal frequency division multiplexing (OFDM) and frequency hopping (FH) are employed [49], and the RF signals must meet the 119 ISM limitations [50]. The most common frequency bands for these signals are at 2.4 GHz and 5 GHz. Figure 1 illustrates the experimental setup for UAV detection. RF signals are acquired using the receiver and stored in a wireless personal digital assistant (PDA). A UAV signal in the 2.5 GHz band is shown in the figure. The main receiver parameters are given in Table 1. The antenna rotates over an and angle in the range 0°–180° at a speed of 22.5°/8 s. The received RF signals are shown in Figure 1 as a matrix where the *x*-axis denotes frequency *f* in the range 2.4 GHz–2.5 GHz and the *y*-axis denotes the RF signal power *y*(*f*) in dBm. This is an *M* × *N* matrix **A** where *M* denotes the number of frequency values and *N* denotes the number of azimuth angle values. The bandwidth of the UAV RF signals is approximately 9.8 MHz [35] so the sampling frequency is *f_w_* = 19.53 kHz.

The device employed in the experiments is a Phantom 4 UAV (DJI, Shenzhen, China) which weighs about 400 g and communicates with the controller using an RF signal at 2.4 GHz. All experiments were conducted in a real environment outdoors at Ocean University of China where there were other devices that share the 2.4 GHz band. Figure 2 shows the measurement locations for the experiments. In the first experiment, the UAV hovered at a height of 100 m at distances of 500 m, 1500 m, and 2400 m from the receiver. This experiment was used for UAV detection using the proposed algorithm. In the second experiment, the UAV was at a distance of 2500 m and 2800 m from the receiver with several interference signals having amplitudes larger than the UAV signal. This experiment was used to test the noise and interference suppression performance of the proposed algorithm. Two UAVs were considered in the third experiment at distances of 2400 m and 2500 m from the receiver as shown in Figure 2b.

## 3. Proposed Method

Figure 3 gives a flowchart of the proposed detection algorithm. In this algorithm, the static and non-static clutter are suppressed using background filtering and linear trend suppression (LTS). The noise is reduced using an automatic gain control (AGC) technique, and the signal to noise ratio (SNR) is improved using an SVD algorithm. Spectrum accumulation (SA) and statistical fingerprint analysis (SFA) methods are employed to provide two frequency estimates of the RF signals. These estimates are used to determine if a UAV is present in the detection environment. A region of interest (ROI) containing the azimuth information is defined using these estimates to reduce the data size and improve the system efficiency. In this section, RF signals acquired at a distance of 500 m between the UAV and receiver are used to illustrate the proposed method.

### 3.1. Clutter Elimination

RF signals are often corrupted by static clutter (interference) with large amplitudes. Further, non-static clutter is introduced by interferers moving in the proximity of the receiver. Thus, techniques to remove this clutter are presented here. The background estimation method is employed to remove the static clutter which is given by [51]:(1)$$B=\frac{1}{M\times N}\sum_{m=1}^{M}\sum_{n=1}^{N}A\left(m,n\right)$$
and this is subtracted from each value in **A** giving the matrix **C**. The LTS algorithm is used to suppress the non-static clutter. The result is [52]:(2)$$D={\left[C^{Τ}-X{\left(X^{Τ}X\right)}^{-1}X^{Τ}C^{T}\right]}^{T}$$
where:(3)$$X=\left[x_{1}\;x_{2}\right]=\left[\begin{array}{cc} 0 & 1\\ 1 & 1\\ \vdots & \vdots \\ N-1 & 1 \end{array}\right]$$

### 3.2. Signal Improvement

The AGC method has been extensively used to enhance weak RF signals. The gain is [53]:(4)$$g_{\mathrm{mask}}=\left\{ \begin{array}{ll} g_{\max} & g_{\mathrm{norm}}\left(i,n\right)\geq g_{\max}\\ g_{\mathrm{norm}}\left(i,n\right) & g_{\mathrm{norm}}\left(i,n\right)<g_{\max} \end{array}\right.$$
where n=0,1,⋯,N−1, i=0,1,⋯,M−ξ, *g*_max_ is the maximum gain, and *g*_norm_(*i*,*n*) is the normalized gain given by:(5)$$g_{\mathrm{norm}}\left(i,n\right)=g\left(i,n\right)/g_{\min}\left(i,n\right)$$
where *g*_min_(*i*, *n*) is the minimum gain for each value of *n* which can be expressed as:(6)g(i,n)=w/e(i,n)
and *e*(*i*, *n*) is the power of the signal in a window of length *ξ* given by:(7)$$e\left(i,n\right)=\sqrt{\sum_{k=i}^{w+i-1}D{\left(i,n\right)}^{2}}$$

Using AGC, *g*_max_ can be predetermined using the calculated gain values. In this paper, *g*_max_ = 0.2. From (4), the *M* × *N* azimuth -frequency matrix **E** is:(8)$$E=D\times g_{\mathrm{mask}}$$

### 3.3. SNR Improvement

SVD is used to improve the SNR. The SVD of (8) can be expressed as [53]:(9)$$\begin{array}{ll} E & =USV^{T}\\ & =\sigma_{1}\left(\begin{array}{c} \cdot \\ \cdot \\ \mu_{1}\\ \cdot \\ \cdot \end{array}\right)\left(\upsilon_{1}\ldots \right)+\sigma_{k}\left(\begin{array}{c} \cdot \\ \cdot \\ \mu_{k}\\ \cdot \\ \cdot \end{array}\right)\left(\ldots \upsilon_{k}\ldots \right)+\cdot \cdot \cdot +\sigma_{N}\left(\begin{array}{l} \cdot \\ \cdot \\ \mu_{N}\\ \cdot \\ \cdot \end{array}\right)\left(\ldots \upsilon_{N}\ldots \right)\\ & =M_{1}+M_{2}+\cdot \cdot \cdot +M_{k}+\cdot \cdot \cdot +M_{N} \end{array}$$
where T denotes matrix transpose, **S** is a diagonal matrix, **U***_M_*
_×_
*_M_* and **V***_N_*
_×_
*_N_* are unitary matrices, and **M***_k_* is the *k*th intrinsic image with the same dimensions as **E**. The singular values *σ_i_* in **S** satisfy $\sigma_{1}^{}\geq \sigma_{2}\geq \sigma_{3}\geq \cdot \cdot \cdot \geq \sigma_{r}\geq 0$. Equation (9) can be expressed as:(10)$$E=M_{\mathrm{UAV}}+M_{\mathrm{noise}}$$
where **M**_UAV_ is the effective RF signal and **M**_noise_ is noise. To remove the noise, the singular values for *i* > 20 are set to zero [54]. The resulting *M* × *N* azimuth -frequency matrix **F** is then [55]:(11)$$F=\sum_{i=1}^{20}u_{i}\sigma_{i}v_{i}^{T}$$

To further improve the SNR, *Q* points in the azimuth dimension are averaged giving [56]:(12)$$G\left(\lambda ,n\right)=\frac{1}{Q}\sum_{m=\lambda Q}^{\left(\lambda +1\right)\;Q-1}F\left(m,n\right)$$
where λ=1,⋯,ℵ. ℵ=⌊M/Q⌋ is the largest integer less than M/Q [57]. **G** is then an ℵ×N azimuth -frequency matrix. In this paper, *Q* = 12. Note that the result in (12) must satisfy the Nyquist sampling theorem. Further, using (12) not only increases the SNR but also reduces the data size which improves the efficiency.

### 3.4. Spectrum Accumulation

An SA algorithm is proposed to obtain a frequency estimate. First, the ℵ×1 matrix **I** is obtained which is given by:(13)$$I\left(\lambda \right)=\sum_{i=1}^{N}{\left|G\left(\lambda ,i\right)\right|}^{2},\;\lambda =1,\cdots ,ℵ$$

To improve the SNR, the envelope:(14)$$P={\left|H(I)\right|}^{2}$$
is used where H() denotes the Hilbert transform. The frequency estimate is then:(15)β=w(α)
where:(16)$$\alpha =\underset{\lambda =1,\cdots ,ℵ}{\arg\max}\left\{P\left(\lambda \right)\right\}$$
is the index of the peak in **P**. *ω* denotes a value in the frequency range 2.4 GHz–2.5 GHz. Figure 4 shows the frequency estimation results using the SA method for the acquired RF signals.

### 3.5. Statistical Fingerprint Analysis

In this section, a frequency estimation method is given which is based on statistical features of the UAV signals. These features include skewness [58], kurtosis [59], and standard deviation [60]. The goal is to detect the presence of a UAV in a given surveillance area. Figure 5 shows these features for the received RF signals at a distance of 500 m. This indicates that the standard deviation is better able to identify UAV signals. Thus, the frequency estimate is:(17)τ=w(υ)
where:(18)$$\upsilon =\underset{j}{\arg\max}\left\{\Phi \left(j\right)\right\}$$
(19)$$\Phi \left[j\right]={\left|H\left(\Psi \left(j\right)\right)\right|}^{2}$$
and Ψ denotes the standard deviation of the values in each row of **I**. *υ* is the index of the peak in Ψ. Figure 6 shows the frequency estimation results using the SFA method with the acquired RF signals.

### 3.6. UAV Determination

As mentioned previously, the bandwidth of the RF signal is 9.8 MHz. This can be used as a threshold to determine if a UAV is present in the detection environment. The error between *τ* obtained using the SA method and *β* obtained using the SFA method is given by: (20)δ=|τ−β|

A UAV is assumed to be present in the detection environment if *δ* ≤ 9.8 MHz.

### 3.7. Data Size Reduction

To disable an unauthorized UAV in a private or sensitive area, the UAV location must be determined. Thus in this section, a ROI is obtained to improve the system efficiency, reduce the data size, and increase the estimation accuracy. As the bandwidth of the RF signals is 9.8 MHz, they are sampled at a frequency of 19.53 kHz, so an area in the range [*τ*, *τ* + 500] is defined as the ROI. The result of using the ROI is illustrated in Figure 7 using the data acquired at a distance of 500 m. The upper figure shows the received RF signals before the ROI is used and the lower figure afterwards. These results indicate that this approach significantly reduces the noise and interference.

### 3.8. Azimuth Estimation

To estimate the azimuth between the UAV and receiver, only the signals in the ROI are considered. The azimuth can be expressed as:(21)γ=η(κ)
where:(22)$$\kappa =\underset{\varepsilon }{\arg\max}\left\{Τ\left(\varepsilon \right)\right\}$$
(23)$$Τ\left(\varepsilon \right)={\left|H\left(Κ\left(\varepsilon \right)\right)\right|}^{2}$$
**K** denotes the standard deviation of the signals in the ROI in the azimuth direction and *κ* is the index of the peak in **K**.

## 4. Results and Discussion

This section presents performance results using the data obtained in the experiments. The proposed algorithm is compared with three well-known methods in the literature such as the ANN [35], constant false alarm rate (CFAR) [56], and higher-order cumulant (HOC) [57] methods. Each dataset contains the RF signals acquired during one rotation of the receive antenna. In the experiments, the starting frequency of the RF signals differs slightly due to the distance between the UAV and receiver. At distances of 500, 1500, 2400, 2500, and 2800 m, the starting frequencies are 2.4225, 2.4225, 2.4681, 2.4225 and 2.4225 GHz, respectively.

### 4.1. Clutter Elimination

Clutter elimination is examined using the RF signals acquired at distances of 500 m and 2800 m. The results for the proposed method are given in Figure 8.

Figure 8a shows the results after spectrum sensing which indicate that the RF signals are too weak to extract due to noise and large amplitude clutter. Background filtering effectively suppresses the clutter as shown in Figure 8b. The results using the AGC method are given in Figure 8c and show increased noise suppression. The results after using LTS in Figure 8d indicate an improvement in the RF signal. Figure 8e,f give the results after the SVD algorithm and the average filter, respectively. Comparing Figure 8a,f confirms that the proposed method significantly reduces the noise and interference. The SNR improvement using the proposed method is evaluated using the received RF signals at different distances and is compared with the corresponding improvement using the ANN and acoustic array methods. The SNR is defined as: (24)$$\mathrm{SNR}=20\log_{10}\left(\sum_{j=1}^{N}\sum_{i=\tau }^{\tau +500}\left|G\left(i,j\right)\right|/\sum_{j=1}^{N}\sum_{i=1}^{M}\left|G\left(i,j\right)\right|\right)$$

The SNR using the proposed method is −11.39 dB for 500 m, −14.57 dB for 1500 m, −22.46 dB for 2400 m, −23.48 dB for 2500 m, and −27.93 dB for 2800 m. These results show that the power of the received signal decreases with distance, as expected. For 500 m, The SNR is −22.89 dB using the ANN method and −23.56 dB using the acoustic array method. Thus, these methods have much lower SNRs at 500 m which indicates poor UAV detection performance. The results for the proposed algorithm are shown in Figure 9a for 1500 m and Figure 9b for 2400 m. These show that the RF signals are improved significantly and the noise is suppressed.

### 4.2. Detection Performance in Strong Interference

The performance of the proposed method in strong interference is evaluated using the signals acquired at distances of 2500 m and 2800 m.

Figure 10 shows that there are several RF interferers in the detection environments. The circular regions denote interference signals with larger amplitudes than the UAV signal. Thus, this is challenging detection problem. The results using the proposed method are given in Figure 11. This shows that the interference is effectively suppressed which will improve the detection performance. The SNR for the proposed, ANN, CFAR, and HOC methods is given in Table 2. This shows that the proposed method provides the best SNR improvement while the ANN method has the worst improvement.

### 4.3. Frequency and Azimuth Estimation

This section examines the frequency and azimuth estimation performance of the proposed method. The frequency estimation results using the SA and SFA algorithms with the data acquired at a distance of 2500 m are shown in Figure 12. Figure 13 shows the azimuth estimation in the ROI using SA for five distances. These results indicate that the UAV azimuth can be acquired with high accuracy using the ROI. The frequency estimates are in the range 2.42 GHz–2.43 GHz when the UAV is at distances of 500 m, 1500 m, 2500 m, and 2800 from the receiver. When the distance between the receiver and UAV is 2400 m, the frequency estimate is in the range 2.46 GHz–2.47 GHz. Further, the bandwidth of the RF signals at these distances is about 10 MHz. These results indicate that the proposed method provides good frequency estimation over a range of distances. The frequency estimation in the ROI using SA is shown in Figure 14 for five distances. Figure 15 shows the frequency estimation in the ROI using SFA for five distances. The frequency estimates *τ* using the SA method are 2.422 GHz for 500 m, 2.422 GHz for 1500 m, 2.468 GHz for 2400 m, 2.422 GHz for 2500 m, and 2.422 GHz for 2800 m, and the frequency estimates *β* using the SFA algorithm are 2.422 GHz for 500 m, 2.422 GHz for 1500 m, 2.468 GHz for 2400 m, 2.422 GHz for 2500 m, and 2.422 GHz for 2800 m. The accuracy of the frequency estimates using different algorithms is given in Table 3. These results indicate that the proposed algorithm provides the highest accuracy.

The azimuth estimation in the ROI using SFA is shown in Figure 16 for five distances. The azimuth estimates with the proposed algorithm are 101° for 500 m, 101° for 1500 m, 110° for 2500 m, 110° for 2800 m, and 67° for 2400 m. Table 4 gives the azimuth estimation accuracy with three methods. These results indicate the proposed method has the smallest error. The detection rates using three algorithms are given in Table 5 and show that better UAV detection performance is obtained with the proposed algorithm.

### 4.4. Detection of Multiple UAVs

The detection of multiple UAVs with the proposed algorithm was also evaluated. Two UAVs were located in the detection environment at distances of 2400 m and 2500 m from the receiver. The received RF signals shown in Figure 17a,b give the results after noise suppression using the proposed method. This clearly indicates two areas where UAVs are present. The frequency estimates using the SA and SFA algorithms are shown in Figure 18, and the corresponding azimuth estimation results are given in Figure 19. The azimuth estimates are 90°–20° and 50°–80° for the two UAVs. The azimuth and frequency estimates for four algorithms are presented in Table 6. With the proposed algorithm, the frequency estimates τ are 2.463 GHz (2400 m) and 2.427 GHz (2500 m), and β are 2.463 GHz (2400 m) and 2.427 GHz (2500 m). The corresponding azimuth estimates are 65° (2400 m) and 112° (2500 m). With the ANN method, the frequency estimates are 2.446 GHz (2400 m) and 2.412 GHz (2500 m), and the azimuth estimates are 30° (2400 m) and 76° (at 2500 m). With the HOC method, the frequency estimates are 2.439 GHz (2400 m) and 2.453 GHz (2500 m), and the azimuth estimates are 22° (2400 m), and 38^°^ (2500 m) HOC. With the CFAR method, the frequency estimates are 2.436 GHz (2400 m) and 2.484 GHz (2500 m), and the azimuth estimates are 37° (2400 m) and 70^°^ (2500 m). These results clearly show the effectiveness of the proposed algorithm as it provides the highest accuracy frequency and azimuth estimates.

## 5. Conclusions

UAV detection is very important due to the threats they can pose to personal and public privacy and safety. In this paper, a method was proposed which employs background filtering to suppress static clutter, and non-static clutter was reduced using linear trend suppression (LTS). Further, automatic gain control (AGC) was used to remove noise and improve the SNR, and the SNR was further increased using singular value decomposition (SVD). The spectrum accumulation (SA) and statistical fingerprint analysis (SFA) methods were employed to obtain frequency estimates. A region of interest (ROI) was defined using these estimates to improve the system efficiency and provide accurate azimuth estimates. To validate the proposed algorithm, experiments were conducted in a real outdoor environment, and the results obtained indicate that it provides excellent UAV detection performance. The recognition rate with this method was near 100% within a distance of 2400 m and greater than 90% within a distance of 3000 m, which is better than with other well-known detection algorithms in the literature. Further, it was shown that the azimuth and frequency of multiple UAVs can be accurately estimated using the proposed algorithm.

## Figures and Tables

**Figure 1 sensors-19-00274-f001:**
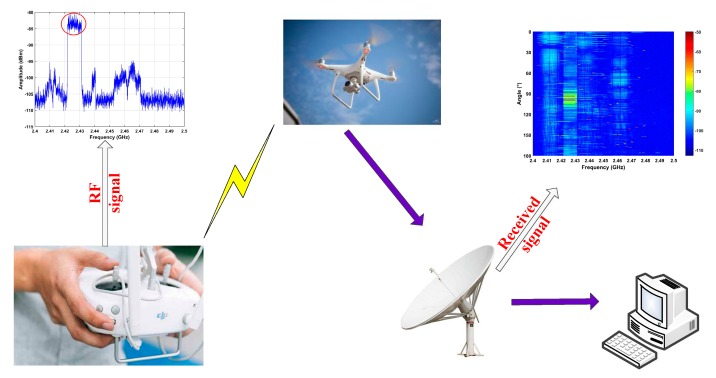
The system model for UAV detection including the receiver, UAV, computer and controller. The receiver is used to collect the RF signals and the computer is used for signal processing. The bidirectional arrow denotes communications between the UAV and controller.

**Figure 2 sensors-19-00274-f002:**
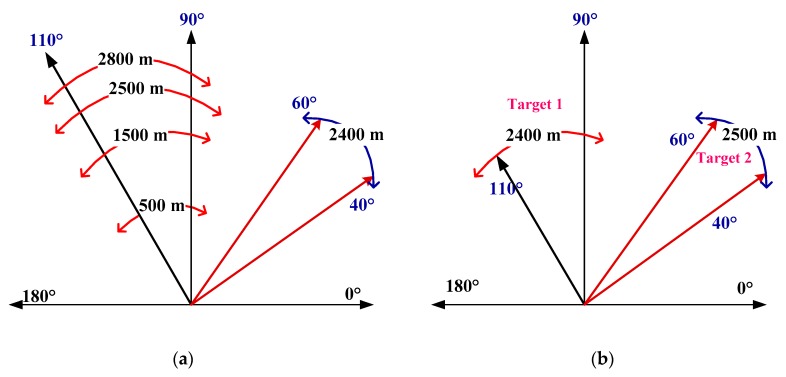
The experiment locations (**a**) UAV hovering at a height of 100 m and 500 m, 1500 m, 2400 m, 2500 m, and 2800 m from the receiver, and (**b**) two UAVs at distances of 2400 m and 2500 m from the receiver.

**Figure 3 sensors-19-00274-f003:**
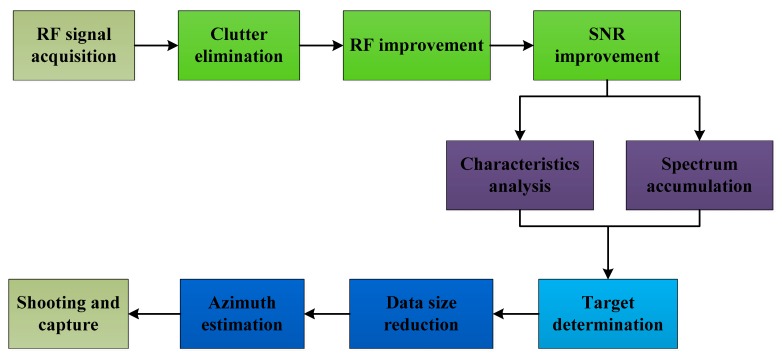
Flowchart of the proposed UAV detection method.

**Figure 4 sensors-19-00274-f004:**
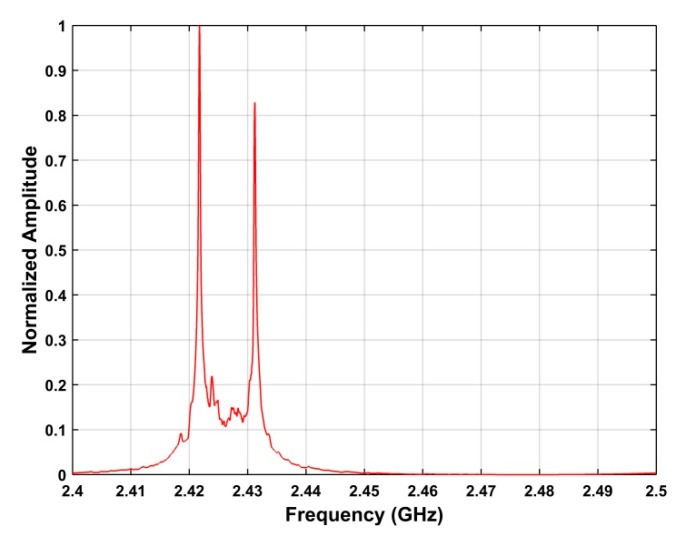
Frequency estimation results using the SA method for the RF signals acquired at a distance of 500 m between the UAV and receiver.

**Figure 5 sensors-19-00274-f005:**
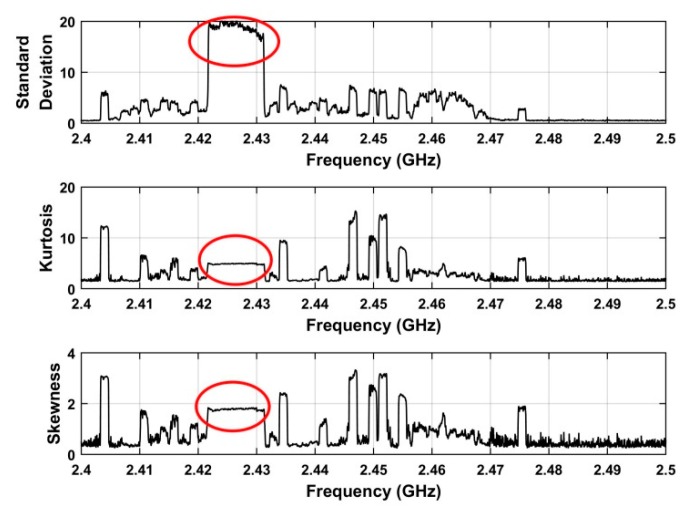
The features extracted from the RF signal acquired at a distance of 500 m between the UAV and receiver.

**Figure 6 sensors-19-00274-f006:**
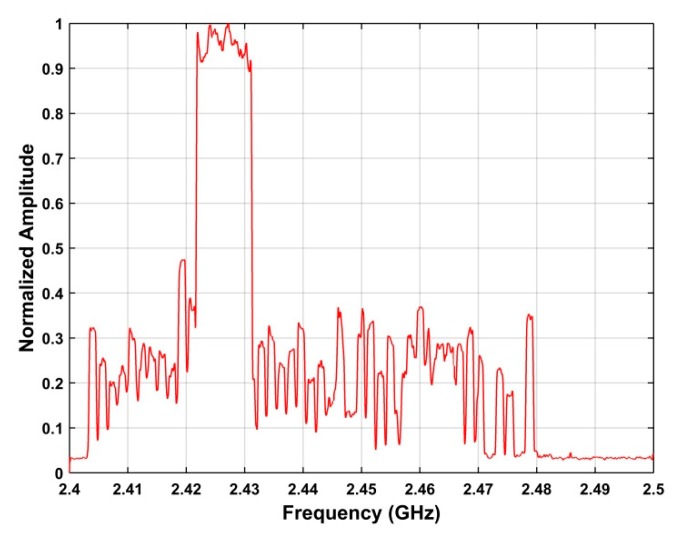
Frequency estimation results using the SFA method with RF signals acquired at a distance of 500 m between the UAV and receiver.

**Figure 7 sensors-19-00274-f007:**
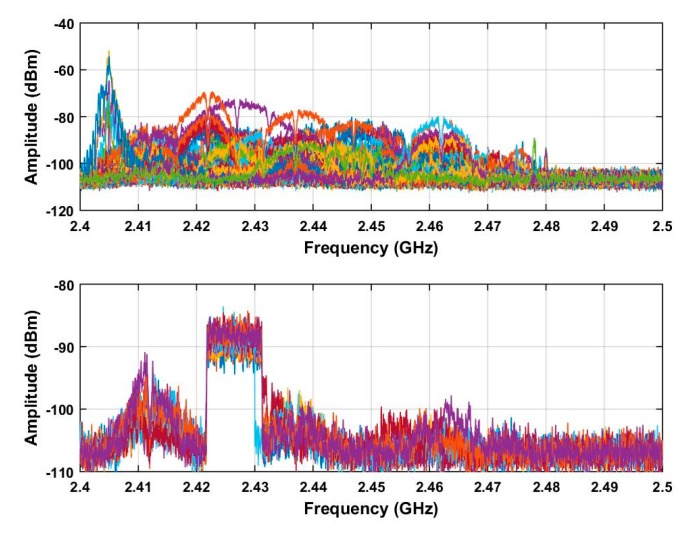
The received RF signals acquired at a distance of 500 m between the UAV and receiver. The upper figure shows all received RF signals and the lower figure shows the signals in the ROI.

**Figure 8 sensors-19-00274-f008:**
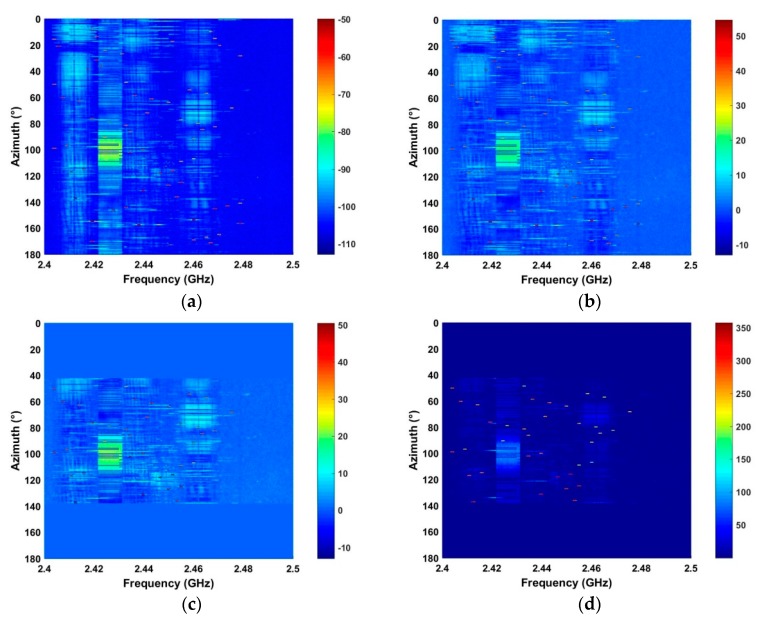
(**a**) The received RF signals, and the results after (**b**) background filtering, (**c**) AGC, (**d**) LTS, (**e**) SVD, and (**f**) average filtering.

**Figure 9 sensors-19-00274-f009:**
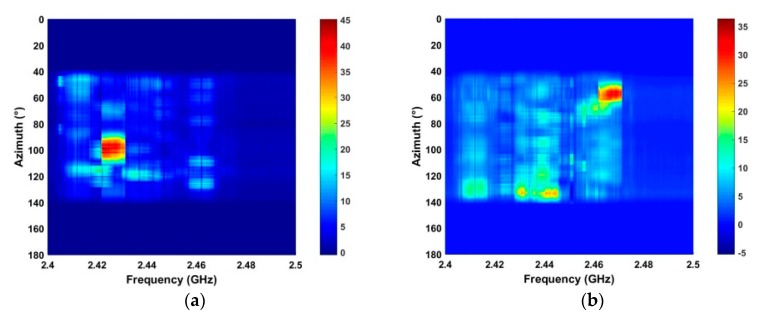
The detection results using the proposed algorithm based on the RF signals acquired at distances of (**a**) 1500 m, and (**b**) 2400 m.

**Figure 10 sensors-19-00274-f010:**
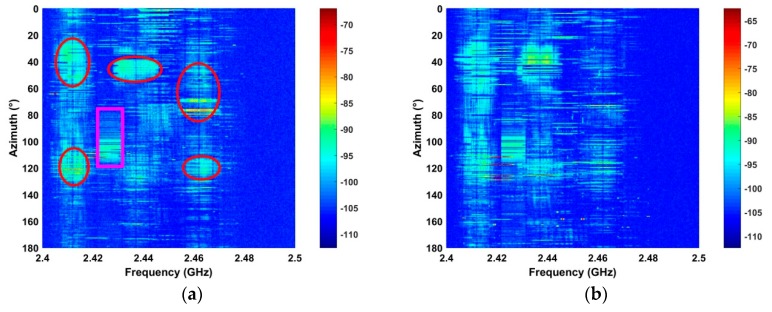
The RF signals acquired in strong interference at distances of (**a**) 2500 m, and (**b**) 2800 m.

**Figure 11 sensors-19-00274-f011:**
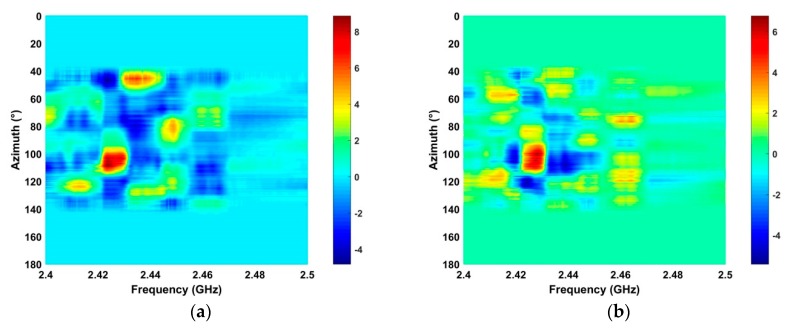
The results for the proposed algorithm using RF signals acquired at a distance of (**a**) 2500 m, and (**b**) 2800 m.

**Figure 12 sensors-19-00274-f012:**
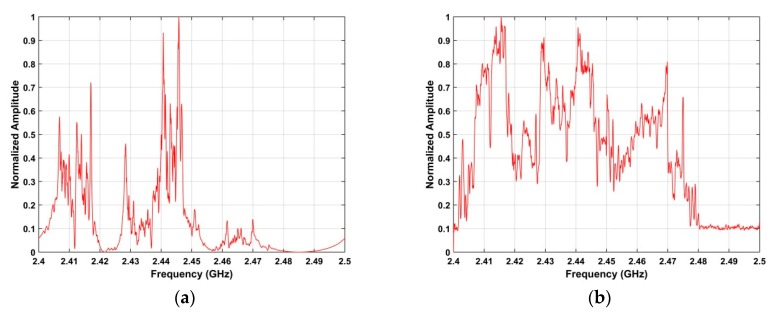
Frequency estimation results using the (**a**) SA, and (**b**) SFA methods.

**Figure 13 sensors-19-00274-f013:**
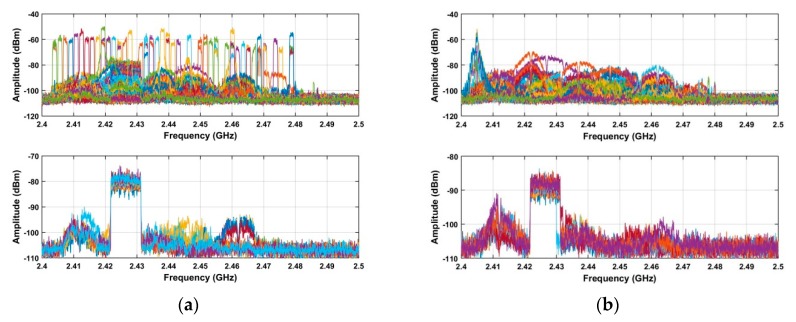
RF signals in the ROI at a distance of (**a**) 500 m, (**b**) 1500 m, (**c**) 2500 m, (**d**) 2800 m, and (**e**) 2400 m.

**Figure 14 sensors-19-00274-f014:**
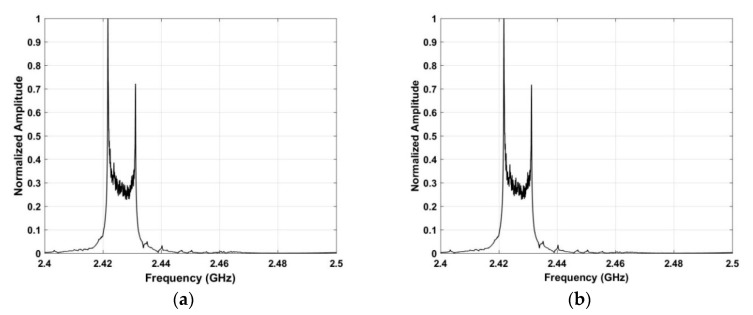
Frequency estimation in the ROI using SA at distances of (**a**) 500 m, (**b**) 1500 m, (**c**) 2500 m, (**d**) 2800 m, and (**e**) 2400 m.

**Figure 15 sensors-19-00274-f015:**
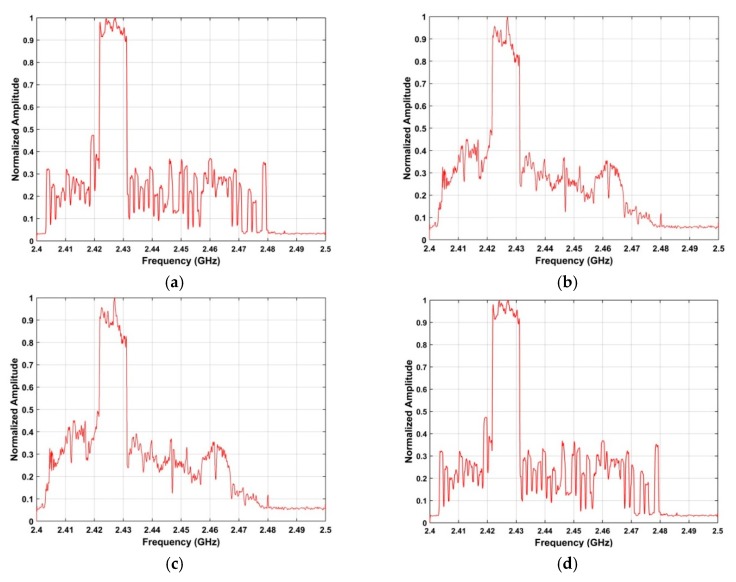
Frequency estimation in the ROI using SFA at distances of (**a**) 500 m, (**b**) 1500 m, (**c**) 2500 m, (**d**) 2800 m, and (**e**) 2400 m.

**Figure 16 sensors-19-00274-f016:**
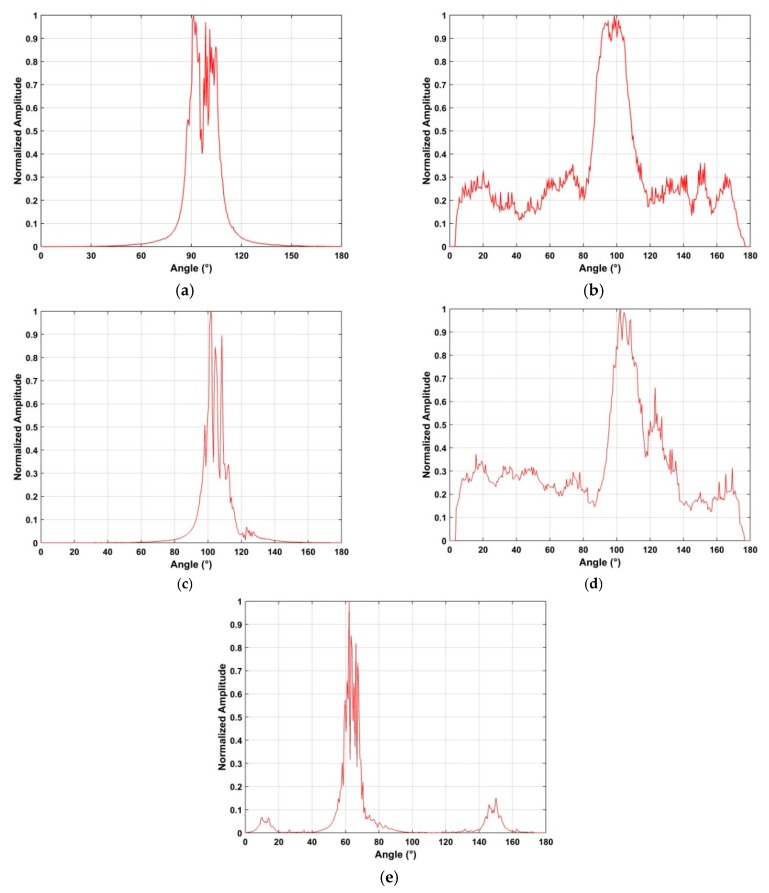
Azimuth estimation in the ROI using SFA at distances of (**a**) 500 m, (**b**) 1500 m, (**c**) 2500 m, (**d**) 2800 m, and (**e**) 2400 m.

**Figure 17 sensors-19-00274-f017:**
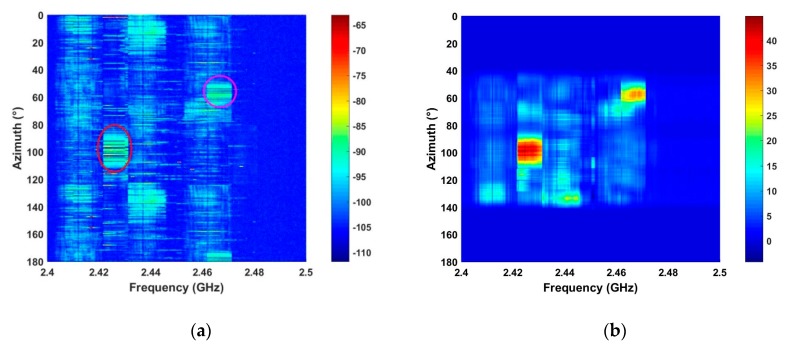
(**a**) The received RF signals and (**b**) the results using the proposed method, with two UAVs at distances of 2400 m and 2500 m.

**Figure 18 sensors-19-00274-f018:**
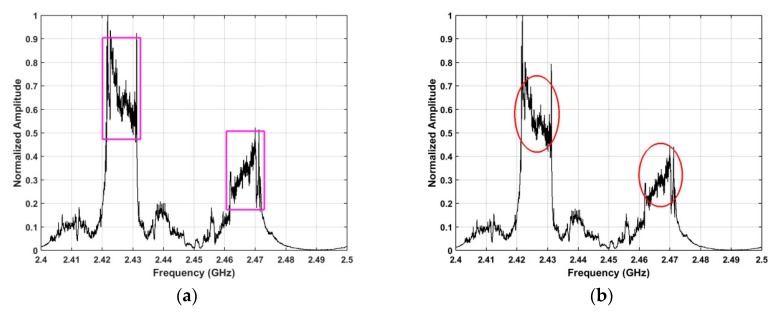
Frequency estimates using the (**a**) SA, and (**b**) SFA methods, with two UAVs at distances of 2400 m 2500 m.

**Figure 19 sensors-19-00274-f019:**
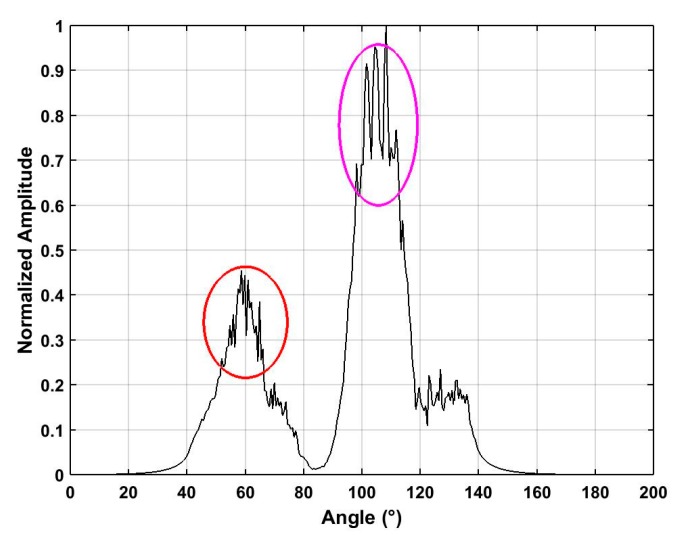
Azimuth estimates using the SFA method with UAVs at distances of 2400 m and 2500 m.

**Table 1 sensors-19-00274-t001:** The main receiver parameters.

Parameter	Value
Gain	24 dBi
Beamwidth	10°
Frequency range	2.3 GHz–2.7 GHz
Azimuth angle	0°–180°
receiver dynamic range	72 dB

**Table 2 sensors-19-00274-t002:** SNR (dB) using four methods with the data obtained at distances of 2500 m and 2800 m. ANN: artificial neural network, CFAR: constant false alarm rate, HOC: higher-order cumulant.

Method	2500 m	2800 m
Proposed	−25.72	−29.71
ANN	−38.35	−43.62
HOC	−31.46	−35.72
CFAR	−29.57	−33.78

**Table 3 sensors-19-00274-t003:** Frequency estimation accuracy with three methods. CFAR: constant false alarm rate, HOC: higher-order cumulant.

Method		Error				
		500 m (MHz)	1500 m (MHz)	2400 m (MHz)	2500 m (MHz)	2800 m (MHz)
Proposed	*τ*	0.5	0.5	0.1	0.5	0.5
	*β*	0.5	0.5	0.1	0.5	0.5
HOC		2.3	5.7	18	15	37
CFAR		4.5	19	27	69	91

**Table 4 sensors-19-00274-t004:** Azimuth estimation accuracy using three methods. CFAR: constant false alarm rate, HOC: higher-order cumulant.

Method	Error (°)				
	500 m	1500 m	2400 m	2500 m	2800 m
Proposed	3.86	5.18	3.35	4.27	7.24
HOC	11.25	19.37	34.96	8.39	12.49
CFAR	7.68	9.24	13.86	9.27	11.24

**Table 5 sensors-19-00274-t005:** Detection rate (%) using three methods. CFAR: constant false alarm rate, HOC: higher-order cumulant.

Method	500 m	1500 m	2400 m	2500 m	2800 m
Proposed method	100	100	100	90	90
HOC	100	90	60	50	50
CFAR	90	90	70	40	40

**Table 6 sensors-19-00274-t006:** Parameter estimates for two UAVs using four different methods. ANN: artificial neural network, CFAR: constant false alarm rate, HOC: higher-order cumulant.

Method	Parameter	2400 m		2500 m	
		Estimate	Error	Estimate	Error
Proposed	Frequency (GHz) τ	2.463	0.005	2.427	0.005
	Frequency (GHz) β	2.463	0.005	2.427	0.005
	Azimuth	65°	5°	112°	12°
ANN	Frequency (GHz)	2.446	0.022	2.453	0.031
	Azimuth	30°	30°	76°	34°
HOC	Frequency (GHz)	2.439	0.029	2.412	0.120
	Azimuth	22°	38°	56°	54°
CFAR	Frequency (GHz)	2.436	0.102	2.484	0.062
	Azimuth	37°	23°	70°	40°

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
