# Peer review of "An Improved Unauthorized Unmanned Aerial Vehicle Detection Algorithm Using Radiofrequency-Based Statistical Fingerprint Analysis"

_sensors, 2019, doi:10.3390/s19020274_

Reviewer 1 Report

The paper presents method for the drone detection and the results are validated by the measurements. The reviewer has the following comments to the authors:

--It is not really clear from the abstract how your method works. Please, provide better explanation in the abstract and include some results already there.

--page 1, line 43: What do the authors mean by higher spatial resolution?

--page 2, lines 59-87: The authors provide a lot of references but give very brief description. It looks just like a list of papers. Instead, it would be beneficial to provide more detailed description of some of them and compare with your work. What are the main advantages?

--page 2, line 89-91: Any evidence that passive radar is better than the active one? In which cases?

--What is the main reason of presenting Table 1? The authors only study one model of a drone and the table is not really relevant to the content of the paper. In addition, drones from completely different segments are mentioned. For example, how drone with the weight of 100 gr can be dangerous to the people? Why do we need to detect this drone? How size of the drone will affect your results?

--Please provide explanation what is passive radar?

--Section 3: Please, provide more detailed description of the proposed detection algorithm. Text description of the diagram will improve the section 3.

--Please, provide more information on fingerprint analysis. Brief description must be presented so that the reader can understand the idea from this paper and not search for the references.

--There are relatively many formulas. Table with all the variables and their meaning will improve the manuscript.

--page 7, line 193: What kind of RF signals? Are they from the drone or from other sources?

--Fig. 4: legend? Otherwise, it can be said in the text what are the curves in this Fig.

--Please, improve the description of section 3.8.

--Why Phantom 4 was used?

--What radar is used?

--Fig. 5 shows the measurement locations rather than the setup.

--Do not divide the figures and do not put them on different pages (Fig. 6).

--Fig. 6 (f): What do the other visible "regions" in this figure mean? They are not visible in Fig. 6 (a). Please provide explanation here.

--page 9, line 241: What is ANN method? Provide the method description in the text.

--Fig. 7: It is not really necessary to show the intermediate steps here since they are given in Fig. 6. The reviewer suggests to delete Fig. 7 (b)-(e).

--What is the acoustic array?

--What do the authors mean by "one drone" and "non-drone"?

--The detailed description of Fig. 8 is missing.

--What is presented in Fig. 9?

--The ratio between the figures and the text is not optimal. Currently, there are too many figures and very little description.

--What do the Figs. 10-11 show? Are they necessary?

--The fonts in Fig. 12 and 13 are not visible. This figures are not clear to the reader and should be explained. What is the reason for presenting them here? What information they give to the reader?

--The conclusion of the paper is very brief and do not provide enough information. If we consider other methods for drone detection - what is the advantage of using the proposed method? Any numbers in percentage for comparison?

--What about failures? The authors should consider also the options where the drones can not be detected. What happens for the smaller drone? Is it still possible to detect? Does the material has any effect?

--Is it possible for the authors to perform any simulations?

Reviewer 3 Report

The authors propose and demonstrate the effectiveness of a detection technique for amateur-grade drones based on the analysis of the continuous drone-ground station communications.

The title seems too unbalanced towards experimentation.  It should reflect better the article even weight between description of the proposed method and experimental validation.

The novelty proposed is not clear.  Is it only experimental or also in processing method?  This should be made clear in text since most of the processing appears to come from external sources.

Description of the proposed method (Section 3) severely lacks relevant details and discussions.  For instance, include in text the rationale behind using a "frequency-azimuth" matrix (line 144), clarify in text what options were considered and why the receipt on lines 148–149 was selected, explanation and demonstration for (2–23), and so on for most decisions, transformations and formulas used in Section 3. The presentation of the method should allow readers to understand it and the rationales behind its components without mandatory consultation of external references (such as [53], [58], [59]).

Clarify in text how "angle" is obtained (e.g., Figure 7 and accompanying text and elsewhere in the manuscript).  The same for "azimuth" (Section 3.8) and clarify (21–23) in text.  E.g., are you using directional antennas, or post-processing, …? Provide all specifications needed for independent experimental replication.

Clarify in text how range is determined (e.g., line 226).

Clarify in text the rationale behind selecting 500 m on line 196.

Clarify in text the meaning of "frequency direction" on line 210, and how "direction" is obtained.

Clarify in text the nature of the errors mentioned on line 219 and (20) and the rationale behind the condition on line 221.

'w' used on line 157 is not defined.  'G' and 'T' in (9) are not defined or ambiguous.

On line 170 both U and V are declared "one unitary matrix".

Size of 'E' is not defined on line 172.  Same for 'F' in (11), 'I' i (13), 'P' in (14).

Define the filter referenced on line 179.

Clarify in text why M/12 is important on line 182.

(15) is not clear.  Explain in text what are the terms, their meanings, measurement units, etc.  The same for (17–19).

Clarify in text how you reached the conclusion on line 227.

Please discuss (24) in text and the rationale behind it.

Clarify what is the "proposed" in Table 2, 4, 5 and elsewhere.

Clarify what is the "filter" in caption of Figure 8.

Please provide full statistical qualification for results in Table 3, 4, 5, error reported on line 303 and in general for all experimental results.  E.g., what is the accuracy of frequency measurements that yield "zero" errors?  How many measurements were made, what is their min, max, average, STDEV?  Clarify in text if the frequencies in Table 3 are classes or actual measurement results.

Please discuss results shown in Figure 11.

Please provide units for the color bar in Figure 12.

Please define "clutter" (used, e.g., on line 143) or use a more informative term.

Please define and use consistent notations.  E.g., square brackets seem to be used both to denote matrix contents (2) and matrix size (1).

On lines 37–41, reference [1] covers mostly UAV regulations and is hardly representative for broad current IoT extension, e.g., see [a] below for smart environment, [b] for remote sensing, [c] for pubic security, [d] for smart traffic, [e] for health care, [f] for intelligent city, [g] for emergency services, [h] for industrial control.  Similarly, [i] below compares satellite/UAV/on-site monitoring techniques supporting authors' conclusions on line 53.

Most captions are not informative.  Figures, tables and their captions should be self-explanatory and convey most of the information even extracted from article context.  The authors should not expect the reader to be well acquainted with the text to understand their meaning.  Specific symbols used in figures or tables should also be briefly explained in their captions.

Table 1 should include indications on drone control range.

Figure 1-a should use a legend to explain the meaning of the pictures (more comprehensive symbols can be used instead of hard to distinguish pictures, especially for B/W prints).

Figures 3 and 5 seem identical.  Same for Figures 13-a and 13-b.

Table 6 appears to miss important symbols like E and G.

The manuscript should be carefully proofread to correct several English improper use, spelling, typos and grammar errors.

–--

[a] Ahmed E, Yaqoob I, Gani A, Imran M, Guizani M.  Internet-of-things-based smart environments: state of the art, taxonomy, and open research challenges.  IEEE Wireless Communications.  2016 Oct;23(5):10-6.

[b] Lazarescu MT.  Design of a WSN platform for long-term environmental monitoring for IoT applications.  IEEE Journal on emerging and selected topics in circuits and systems.  2013 Mar;3(1):45-54.

[c] Lau BP, Wijerathne N, Ng BK, Yuen C. Sensor fusion for public space utilization monitoring in a smart city.  IEEE Internet of Things Journal.  2018 Apr;5(2):473-81.

[d] Roy A, Siddiquee J, Datta A, Poddar P, Ganguly G, Bhattacharjee A. Smart traffic & parking management using IoT.  InInformation Technology, Electronics and Mobile Communication Conference (IEMCON), 2016 IEEE 7th Annual 2016 Oct 13 (pp. 1-3).  IEEE.

[e] Islam SR, Kwak D, Kabir MH, Hossain M, Kwak KS. The internet of things for health care: a comprehensive survey. IEEE Access.  2015;3:678-708.

[f] Latre S, Leroux P, Coenen T, Braem B, Ballon P, Demeester P.  City of things: An integrated and multi-technology testbed for IoT smart city experiments.  InSmart Cities Conference (ISC2), 2016 IEEE International 2016 Sep 12 (pp. 1-8).  IEEE.

[g] Dastjerdi AV, Sharifi M, Buyya R. On Application of Ontology and Consensus Theory to Human-Centric IoT: an Emergency Management Case Study.  InData Science and Data Intensive Systems (DSDIS), 2015 IEEE International Conference on 2015 Dec 11 (pp.  636-643).  IEEE.

[h] Sheng Z, Mahapatra C, Zhu C, Leung V. Recent advances in industrial wireless sensor networks towards efficient management in IoT.  IEEE access.  2015 May 19;3:622-37.

[i] Lazarescu MT.  Design and field test of a WSN platform prototype for long-term environmental monitoring.  Sensors.  2015 Apr 22;15(4):9481-518.

Author Response

Reply to the Reviewer of An Improved Unauthorized UAV Detection Algorithm Using RF-based Statistical Fingerprint Analysis

sensors-400979

We would like to thank you for the time spent in evaluating our submission. The reviewer comments have helped tremendously in improving this paper. We hope the revised manuscript will be acceptable. Below you will find our point-by-point responses to all the reviewer comments/questions.

The authors propose and demonstrate the effectiveness of a detection technique for amateur-grade drones based on the analysis of the continuous drone-ground station communications.

The title seems too unbalanced towards experimentation.  It should reflect better the article even weight between description of the proposed method and experimental validation.

The title has been updated.

The novelty proposed is not clear.  Is it only experimental or also in processing method?  This should be made clear in text since most of the processing appears to come from external sources.

The contributions of the paper have been added in Introduction.

Description of the proposed method (Section 3) severely lacks relevant details and discussions. For instance, include in text the rationale behind using a "frequency-azimuth" matrix (line 144), clarify in text what options were considered and why the receipt on lines 148–149 was selected, explanation and demonstration for (2–23), and so on for most decisions, transformations and formulas used in Section 3. The presentation of the method should allow readers to understand it and the rationales behind its components without mandatory consultation of external references (such as [53], [58], [59]).

To make readers easy to read and understand the whole paper, Section 3 has been updated.

In the proposed method, the static and non-static clutters with stronger amplitudes affecting RF signals detection are suppressed based on the background filtering and linear trend suppression (LTS) methods, effectively. The noises are eliminated based on the automatic gain control (AGC) technique, and the signal to noise ratio (SNR) is improved using SVD algorithm. The SA and SFA methods are proposed to provide two frequency estimates of RF signals. These estimates are used to determine if a UAV is present in the detection environment. Based on the frequency estimate, ROI containing the azimuth information is defined to reduce the data size and improve the system efficiency.

Clarify in text how "angle" is obtained (e.g., Figure 7 and accompanying text and elsewhere in the manuscript). The same for "azimuth" (Section 3.8) and clarify (21–23) in text. E.g., are you using directional antennas, or post-processing? Provide all specifications needed for independent experimental replication.

As shown in Figure 1, the antenna rotates within 0o-180o with a speed of 22.5o/ 8 s. RF signals are shown in Figure 1, where the x-axis denotes the frequency f in the range 2.4 GHz-2.5 GHz, and the y-axis denotes the power y(f) of RF signal in dBm. The frequency is sampled with the interval fw = 19.53 kHz. The bandwidth of RF signal is approximately 9.8 MHz. Assuming that the acquired RF signals are referred as AM×N, where M denotes the number of the frequency samples, and N denotes the number of the azimuth angle samples.

Thus, the angle information can be acquired.

Clarify in text how range is determined (e.g., line 226).

Due to the bandwidth of RF signal is 9.8 MHz and the frequency is sampled with the interval fw = 19.53 kHz, the range can be obtained.

Clarify in text the rationale behind selecting 500 m on line 196.

To make readers easy to understand the proposed detection method, RF signals acquired at a distance of 500 m between UAV and the receiver are used as examples to explain the proposed UAV detection method in section 3.

Clarify in text the meaning of "frequency direction" on line 210, and how "direction" is obtained.

In the used receiver, as shown in Figure 1, the RF signals AM×N are acquired, where M denotes the number of the frequency samples, and N denotes the number of the azimuth angle samples. Here the frequency direction denotes the row of the matrix, which has been amended in this paper. 

Clarify in text the nature of the errors mentioned on line 219 and (20) and the rationale behind the condition on line 221.

The bandwidth of RF signal is 9.8 MHz.

Using SA and SFA methods proposed in this paper, two frequency estimates can be acquired. Based on the results, the error between two frequency estimates can be acquired, which are usually less than 9.8 MHz. Thus, in this paper, (20) is used to determine if a UAV is present in the detection environment. And section 4 validate the effective of the used bandwidth.

'w' used on line 157 is not defined.  'G' and 'T' in (9) are not defined or ambiguous.

These errors have been added in this paper.

On line 170 both U and V are declared "one unitary matrix".

U and V are unitary matrix with different dimensions.

Size of 'E' is not defined on line 172. Same for 'F' in (11), 'I' i (13), 'P' in (14).

E F I P all are M×N matrix.

Define the filter referenced on line 179.

This filter has been defined.

 (15) is not clear. Explain in text what are the terms, their meanings, measurement units, etc.

Using (16), a can be acquired, which is the index of the peak in P. Using the index, the frequency estimate can be acquired due to ω is the frequency samples in 2.4 GHz-2.5 GHz.

Clarify in text how you reached the conclusion on line 227.

As shown in Figure 6, the upper figure shows RF signals are covered by noises with larger amplitudes. However, using ROI defined in this paper, the RF signals can be acquired as shown in the following figure, which can remove the noises. Thus, SNR can be improved.

Clarify what is the "proposed" in Table 2, 4, 5 and elsewhere.

All tables have been updated.

Clarify what is the "filter" in caption of Figure 8.

Figure 8 has been updated.

Please discuss results shown in Figure 11.

Figure 11 has been discussed.

Please provide units for the color bar in Figure 12.

Figure 12 has been updated.

Please define "clutter" (used, e.g., on line 143) or use a more informative term.

In this paper, clutter denotes the acquired signals from other objects.

Please define and use consistent notations. E.g., square brackets seem to be used both to denote matrix contents (2) and matrix size (1).

All these questions have been amended.

On lines 37–41, reference [1] covers mostly UAV regulations and is hardly representative for broad current IoT extension, e.g., see [a] below for smart environment, [b] for remote sensing, [c] for pubic security, [d] for smart traffic, [e] for health care, [f] for intelligent city, [g] for emergency services, [h] for industrial control. Similarly, [i] below compares satellite/ UAV/ on-site monitoring techniques supporting authors' conclusions on line 53.

[a] Ahmed E, Yaqoob I, Gani A, Imran M, Guizani M.  Internet-of-things-based smart environments: state of the art, taxonomy, and open research challenges.  IEEE Wireless Communications.  2016 Oct;23(5):10-6.

[b] Lazarescu MT.  Design of a WSN platform for long-term environmental monitoring for IoT applications.  IEEE Journal on emerging and selected topics in circuits and systems.  2013 Mar;3(1):45-54.

[c] Lau BP, Wijerathne N, Ng BK, Yuen C. Sensor fusion for public space utilization monitoring in a smart city.  IEEE Internet of Things Journal.  2018 Apr;5(2):473-81.

[d] Roy A, Siddiquee J, Datta A, Poddar P, Ganguly G, Bhattacharjee A. Smart traffic & parking management using IoT.  InInformation Technology, Electronics and Mobile Communication Conference (IEMCON), 2016 IEEE 7th Annual 2016 Oct 13 (pp. 1-3).  IEEE.

[e] Islam SR, Kwak D, Kabir MH, Hossain M, Kwak KS. The internet of things for health care: a comprehensive survey. IEEE Access.  2015;3:678-708.

[f] Latre S, Leroux P, Coenen T, Braem B, Ballon P, Demeester P.  City of things: An integrated and multi-technology testbed for IoT smart city experiments.  InSmart Cities Conference (ISC2), 2016 IEEE International 2016 Sep 12 (pp. 1-8).  IEEE.

[g] Dastjerdi AV, Sharifi M, Buyya R. On Application of Ontology and Consensus Theory to Human-Centric IoT: an Emergency Management Case Study.  InData Science and Data Intensive Systems (DSDIS), 2015 IEEE International Conference on 2015 Dec 11 (pp.  636-643). IEEE.

[h] Sheng Z, Mahapatra C, Zhu C, Leung V. Recent advances in industrial wireless sensor networks towards efficient management in IoT. IEEE access. 2015 May 19;3:622-37.

[i] Lazarescu MT.  Design and field test of a WSN platform prototype for long-term environmental monitoring. Sensors. 2015 Apr 22; 15(4):9481-518.

All these questions have been amended.

Most captions are not informative. Figures, tables and their captions should be self-explanatory and convey most of the information even extracted from article context.  The authors should not expect the reader to be well acquainted with the text to understand their meaning.  Specific symbols used in figures or tables should also be briefly explained in their captions.

All captions have been updated.

Table 1 should include indications on drone control range.

Figure 1-a should use a legend to explain the meaning of the pictures (more comprehensive symbols can be used instead of hard to distinguish pictures, especially for B/W prints).

Figure 1 has been updated.

Figures 3 and 5 seem identical. Same for Figures 13-a and 13-b.

Figures 3 and 5 have been updated.

Table 6 appears to miss important symbols like E and G.

E is given in (8), and G is given in (12).

The manuscript should be carefully proofread to correct several English improper use, spelling, typos and grammar errors.

The whole paper has been updated and revised. 

Reviewer 4 Report

This paper is proposing an improved radio frequency based method to detect drones using a spectrum sensing system. Based on the RF signals acquired in real-life scenarios, the clutter is eliminated using the background filtering method. The singular value decomposition algorithm and an improved filter have been used to reduce the worse influences of the noises on drone detection, and this can better improve the signal to noise ratio (SNR) of RF signals. The spectrums accumulation (SA) algorithm and the statistical fingerprint analysis (SFA) method have been proposed to provide two frequency estimates of RF signals. The detection results of this study in real-life scenarios have proved the better capability and effectiveness of detecting drones in all conducted experiments, which make it a promising algorithm for drone detection. The results are relevant and I recommend this manuscript for publication in sensor after addressing some minor revisions. It can be accepted, just review some sentences as described below:

Minor comments:

Page 2:

-          Review papers about drones and their application can be cited here.

-          Drone in general can be categorized to different classes (UAV, MAV, NAV, etc.) and types (Fixed wings, flapping wings, multirotors, etc.). Please clarify in the introduction which types and classes of the drones are targeted in this paper.

-          The third paragraph in page 2 is so long. Please break it to two or three paragraphs.

Author Response

Reply to the Reviewer of An Improved Unauthorized UAV Detection Algorithm Using RF-based Statistical Fingerprint Analysis

sensors-400979

We would like to thank you for the time spent in evaluating our submission. The reviewer comments have helped tremendously in improving this paper. We hope the revised manuscript will be acceptable. Below you will find our point-by-point responses to all the reviewer comments/questions.

Minor comments:

Page 2:

- Review papers about drones and their application can be cited here.

The corresponding references have been added in the revised paper.

- Drone in general can be categorized to different classes (UAV, MAV, NAV, etc.) and types (Fixed wings, flapping wings, multirotors, etc.). Please clarify in the introduction which types and classes of the drones are targeted in this paper.

The drone used in this paper has been changed as UAV.

- The third paragraph in page 2 is so long. Please break it to two or three paragraphs.

The third paragraph has been divided into two paragraphs. 

Round  2

Reviewer 1 Report

The manuscript has been significantly improved. The reviewer has minor comments:

--Table 1 is not necessary in this paper. The frequencies and drone sizes can be mentioned in the text. There are much more drone models, so the reviwer proposes just to summarize the key parameters in the text and delete table 1. Moreover, the authors did not answer how their algorithm will work for other drone sizes.

--Passive radar explanation can be given in the text.

--Related to the RF signals used by drones to communicate with the ground control station: does interference take place? How interefering signals affect the results.

Author Response

Reply to the Reviewer of An Improved Unauthorized UAV Detection Algorithm Using RF-based Statistical Fingerprint Analysis

sensors-400979

We would like to thank you for the time spent in evaluating our submission. The reviewer comments have helped tremendously in improving this paper. We hope the revised manuscript will be acceptable. Below you will find our point-by-point responses to all the reviewer comments/questions.

The manuscript has been significantly improved. The reviewer has minor comments:

--Table 1 is not necessary in this paper. The frequencies and drone sizes can be mentioned in the text. There are much more drone models, so the reviewer proposes just to summarize the key parameters in the text and delete table 1. Moreover, the authors did not answer how their algorithm will work for other drone sizes.

Table 1 has been deleted.

--Passive radar explanation can be given in the text.

The passive radar has been changed as spectrum sensing.

--Related to the RF signals used by drones to communicate with the ground control station: does interference take place? How interefering signals affect the results.

RF signals used by UAV to communicate with the ground control station are in 2.4 GHz.

Many devices work in the same frequency band as 2.4 GHz, which have larger amplitudes and make RF signals challenging to detect. 

Reviewer 2 Report

The paper discusses a spectrum sensing and detection method for UAVs based on enhancements techniques used in the radar domain.

The cross connection between techniques used in radar and spectrum sensing is interesting, but as I mentioned in my first review, the novelty of the paper is not enough for a journal publication. I’m convinced that the paper would fit perfectly in a conference on spectrum sensing or on counter-UAV.

The mathematical development and the formulas are still sloppy. In the first review, I pointed out more than 10 errors/typos in the formulas. In the new version of the paper I still found more then 5 inexactitudes. I really have the impression the authors are using the reviewers more as correctors than reviewers.

Some weak points in the paper (in the proposed method):

- Line 180: The automatic (autonomous) selection of the correct value of ’n' is  not an easy task. How do you know how many singular values belong to the drone and how many belong to the noise? I can imagine multiple techniques exist to determine a good value of 'n'. Is this value of 'n' dependent on the type of drone? What is the influence of the SNR in the selection of ’n’?

- Line 206: Experimentally, you observe that the standard deviation is a good feature in the SFA for the Phantom 4 drone and the SNR range of your acquisitions. But is this also for other drones or other SNR ranges?

- Line 222: The test-statistic \delta is not a good  ( or a too naïve) test statistic. It will yield a minimum false alarm rate P_fa of 10% is a scenario when no drone is present.

-Line 223: To simple test: \delta < 9;8 MHz is only valid for a phantom 4. What about other drones ….

Some other remarks, comments:

- §2 system model: I would, in this paragraph mention that you use a radar receiving system to acquire the RF signal, with a directive rotating antenna.

- line 159: n=0,2,..N-1. why n=1 is excluded?

- Line 159, 163 and 164. I suppose you are talking about the same ‘w’. Please use the same symbol…

- Line 168: mismatch of dimensions!. In line 159 you put i=0,1,..M-w. This means that g_mask has dimensions Nx(M-w). D has dimensions NxM

- Line 188: You sum over ‘i’, and the result is still a function of ‘i’. This should be I[m]!

- Line 193 &195: use the same symbol! Either \omega or ‘w’. All this small inexactitudes make it very hard to read the paper.

- Line 212: \nu is the index of the peak in \Phi (envelope of \Psi), see line 210. I’m not convinced that the peak in a function or a peak in the envelope of a function are always the same.

-Line 211. Why do you need to take the envelope? Please explain.

-Line 228: What is \psi ? It’s not explained in the text, neither in table 6. 

- Line 228: How do you get the number \psi+500? Is this specific for a Phantom 4? How can you then generalise your method?

- Line 231: What represents figure 6(a)? How do you get from 6(a) to 6(b)?

- Line 241: same remark as line 212

Author Response

Reply to the Reviewer of An Improved Unauthorized UAV Detection Algorithm Using RF-based Statistical Fingerprint Analysis

sensors-400979

We would like to thank you for the time spent in evaluating our submission. The reviewer comments have helped tremendously in improving this paper. We hope the revised manuscript will be acceptable. Below you will find our point-by-point responses to all the reviewer comments/questions.

The paper discusses a spectrum sensing and detection method for UAVs based on enhancements techniques used in the radar domain. The cross connection between techniques used in radar and spectrum sensing is interesting, but as I mentioned in my first review, the novelty of the paper is not enough for a journal publication. I’m convinced that the paper would fit perfectly in a conference on spectrum sensing or on counter-UAV.

The contributions of the paper can be summarized as:

(1)   The spectrums accumulation (SA) algorithm and statistical fingerprint analysis (SFA) method are proposed to provide frequency estimates of RF signals. These estimates are used to determine if a UAV is present in the detection environment.

(2)   The region of interest (ROI) is defined to reduce the data size, improve the system efficiency and provide accurate azimuth estimate.

The mathematical development and the formulas are still sloppy. In the first review, I pointed out more than 10 errors/typos in the formulas. In the new version of the paper I still found more than 5 inexactitudes. I really have the impression the authors are using the reviewers more as correctors than reviewers.

The whole paper has been updated.

Some weak points in the paper (in the proposed method):

-Line 223: To simple test: \delta < 9;8 MHz is only valid for a phantom 4. What about other drones ….

\delta < 9.8 MHz is valid for the most UAV due to RF has to meet the requirements of 119 ISM 2.4 GHz band.

- §2 system model: I would, in this paragraph mention that you use a radar receiving system to acquire the RF signal, with a directive rotating antenna.

The key parameters of the used receiver are shown in Table 1. The antenna rotates within 0o-180o with a speed of 22.5o/ 8 s. RF signals are shown in Figure 1, where x-axis denotes the frequency f in the range of 2.4 GHz-2.5 GHz, and y-axis denotes the power y(f) of RF signals in dBm. The frequency is sampled with an interval fw = 19.53 kHz. The bandwidth of RF signals is approximately 9.8 MHz. Assuming that the acquired RF signals using the receiver as shown in Figure 1 are referred as a M×N matrix A, where M denotes the number of frequency samples, and N denotes the number of azimuth angle samples.

- line 159: n=0,2,..N-1. why n=1 is excluded?

This error has been changed.

- Line 159, 163 and 164. I suppose you are talking about the same ‘w’. Please use the same symbol…

The errors have been changed.

- Line 188: You sum over ‘i’, and the result is still a function of ‘i’. This should be I[m]!

This error has been changed.

- Line 193 &195: use the same symbol! Either \omega or ‘w’. All this small inexactitudes make it very hard to read the paper.

The error has been changed.

-Line 211. Why do you need to take the envelope? Please explain.

The envelope is used to suppress the noises in the range 2.4 -2.5 GHz.

-Line 228: What is \psi ? It’s not explained in the text, neither in table 6. 

\psi has been changed as τ acquired in (17).

- Line 228: How do you get the number \psi+500? Is this specific for a Phantom 4? How can you then generalise your method?

Due to the bandwidth of RF signal is 9.8 MHz and the frequency is sampled with the interval fw = 19.53 kHz, the range can be obtained.

- Line 231: What represents figure 6(a)? How do you get from 6(a) to 6(b)?

As shown in Figure 6, the upper figure indicates RF signals are covered with noises with larger amplitudes, and RF signals are acquired using ROI as shown in the following figure. Results indicate the noises can better be removed using ROI, which can improve SNR effectively.

- Line 241: same remark as line 212

Line 241 has been updated. 

Reviewer 3 Report

The authors addressed most comments. Some were silently ignored, some were superficially and/or inadequately addressed.

Notwithstanding authors' reassurances and after several review rounds, the article is still poorly presented, lacks scientific rigor and coherence, has sloppy mistakes, missing or too generic explanations on the method and experimental results, unrelated or unexplained references and approximate command of English language.  See below for details.

The authors should make every effort to be clear and specific in their scientific communications out of respect for the scientific community they address, for the volunteered time of the reviewers and, last but not least, because the authors themselves indelibly sign the (eventually published) article, which is then going to publicly represent them and their ability to conduct and present rigorous and good quality research (or the lack thereof).

Full statistical qualification of experimental results should be provided in tables, plots and text.  How many measurements were made, what is their min, max, average, STDEV.

Define clearly in text the sizes of E F I P.

If S is "one diagonal matrix" (line 192) then it has only "1" on the diagonal and cannot have "singular values" σi, as stated on line 193.  Include a representation of S and correct the text.

Clarify in text how gain control (AGC) can "suppress" noise. Amplifier gain affects signal and noise pretty much the same way, hence changing gain does not significantly change SNR. Also, use meaningful references.  For instance, [59], apparently used by authors to exemplify AGC operation and effects on
line 176, does not address AGC and its (supposedly de-noising) effects.

Moreover, clarify in text the types of noise addressed by each method.  The readers are presented:
- "clutter elimination" (Section 3.1, lines 165–166)
- "suppress non-static clutter" (Section 3.1, line 169)
- "suppress noises" (Section 3.2, lines 174–175)
Without proper definitions of "static clutter", "non-static clutter", and "noises" it appears that all possible noise was already "eliminated" or "suppressed" and there is nothing left to improve when Section 3.3 announces some other vaguely-defined  "SNR improvement".

Clarify in text why "12" in "M/12" is important on line 204 and how [58] is relevant in this context.

Plots in Figures 4 and 6 are identical, although they refer to distinct estimation methods.  Include correct plots or properly analyze and explain the identity in the article.

Text in plots in Figure 1 is not readable, but is referenced in text on lines 121–126.  Please increase character size or refer the reader to adequately-sized plots included in the article.

Include measurement units for the numbers on the color bars in Figure 11.

Define all abbreviations at first use in article, e.g., SNR.

The manuscript should be carefully proofread by a proficient English speaker to correct several important English issues.

Author Response

We would like to thank the reviewers for the time spent in evaluating our submission, and the editor for handling our paper. The reviewer comments have helped tremendously in improving this paper. We hope the revised manuscript will be acceptable. Below you will find our point-by-point responses to all the reviewer and editor comments/questions.

Round  3

Reviewer 2 Report

I still have a problem with Figure 7. I have the impression that in fig.7(a), we see three types of signals: 1. a frequency hopping signal, which is most probably the uplink between the RC and the drone (Not a 'noises with larger amplitudes' as the author calls it in the paper), 2. the signal between 2.42 and 2.43 GHz, with bandwidth of 9.8 MHz (is this the downlink signal???, i'm not sure, is it telemetry, is it video, ...), and 3. some WiFi signals (most probably from some APs). The author needs to give more details here.  What is the 9.8 MHz signal, is it telemetry, video, ... If so, is this representative for all types of drones? I know that some drones also use a frequency hopping signal in the downlink, or a classical WiFi signal for the video downlink,....

Some of my previous remarks are completely neglected. E.g.:

- how to find 'n' (line 200),

- \delta is not a good test statistic (line 242), as it will yield a false alarm rate of min 10 % (9.8/80 to be exact). You can not use this \delta to make a decision on the presence of a drone...

Author Response

We would like to thank the reviewers for the time spent in evaluating our submission, and the editor for handling our paper. The reviewer comments have helped tremendously in improving this paper. We hope the revised manuscript will be acceptable. Below you will find our point-by-point responses to all the reviewer and editor comments/questions.

Reviewer 3 Report

The authors addressed sufficiently my comments.

Include the units that you mentioned in reply (dBm) in caption or plots in Figure 11.